# Impact of COVID-19 in Patients with Lung Cancer: A Descriptive Analysis

**DOI:** 10.3390/ijerph20021583

**Published:** 2023-01-15

**Authors:** Damian Mojsak, Michał Dębczyński, Beata Kuklińska, Łukasz Minarowski, Agnieszka Kasiukiewicz, Anna Moniuszko-Malinowska, Piotr Czupryna, Robert Marek Mróz

**Affiliations:** 12nd Department of Lung Diseases and Tuberculosis, Medical University of Białystok,14 Żurawia Street, 15-540 Bialystok, Poland; 2Department of Geriatrics, Medical University of Białystok, 27 Fabryczna Street, 15-471 Bialystok, Poland; 3Department of Infectious Diseases and Neurology, Medical University of Białystok, 14 Żurawia Street, 15-540 Bialystok, Poland

**Keywords:** lung cancer, COVID-19, cancer care, epidemiology

## Abstract

The COVID-19 pandemic poses a challenge to health systems worldwide. Limiting healthcare availability may delay early diagnosis and worsen the treatment effects of various diseases, including oncological diseases. We analyzed patients presenting to the 2nd Department of Lung Diseases and Tuberculosis in Białystok, Poland, with suspicion of lung cancer 12 months prior to the COVID-19 pandemic (pre-COVID-19) and, similarly, 12 months after the outbreak of the pandemic (mid-COVID). In total, 320 patients were analyzed—132 prior to and 188 after the COVID-19 outbreak. During the COVID-19 period, there was a lower percentage of patients presenting with ECOG performance status 0-1, with a noticeably increased percentage of patients with ECOG PS ≥2. The disease’s clinical stage (CS) was higher on admission during COVID-19. We observed more use of immunotherapy and more deaths before the start of treatment during the COVID-19 period. These results provide insight into the early effects of the COVID-19 pandemic on lung cancer patients and underscore the importance of conducting further studies to assess the long-term effects of the COVID-19 pandemic on this population.

## 1. Introduction

The global outbreak of coronavirus disease 2019 (COVID-19) caused by the severe acute respiratory syndrome coronavirus 2 (SARS-CoV-2), declared by the World Health Organization as a global pandemic on the 11 of March 2020, has affected healthcare systems around the world [1,2]. As of 18 December 2022, over 649 million confirmed cases and over 6.6 million deaths have been reported globally [3].

Lung cancer remains the leading cause of cancer-related deaths in men and women with over 2 million cases globally [4,5]. Despite the advances in diagnostics and treatment, the survival of these patients remains relatively short [6]. Quick and early diagnosis and access to treatment is the key factor responsible for improving the prognosis of lung cancer patients [7].

The pandemic has had a huge impact on the functioning of health care [8]. One group particularly affected by the pandemic include cancer patients. This specific group of patients was disproportionately affected by the COVID-19 pandemic with the impact on the delivery of cancer care, adverse treatment outcomes, and even disruption of cancer research [9]. The most commonly considered effect is a direct impact, i.e., illnesses and deaths caused by COVID-19. However, the indirect impact of limited access to healthcare for patients suffering from other diseases is often overlooked [10]. Security measures taken by the government and hospital administration forced physicians to make difficult ethical decisions in order to reduce the risk of SARS-CoV infection, which often led to delays in diagnosis and treatment [11].

By reducing the number of places in oncology departments, hindering access to diagnostics, and interrupting or terminating the treatment regime, the pandemic posed a particular challenge for the cancer population. However, the effect of the pandemic on treatment outcomes has not been well understood so far, and much attention is now being paid to collecting data on the impact of COVID-19 on cancer patients [12,13,14]. Protecting patients with lung cancer from COVID-19 complications, while avoiding treatment delays, has brought unique challenges to healthcare facilities. The symptoms presented by patients with lung cancer, such as cough, dyspnea, and hemoptysis are consistent with those that occur in COVID-19, which additionally hindered proper diagnostics and therapeutic management [15,16,17].

Based on the observation of lung cancer patients reporting to our department, we decided to analyze this population, which included patients reporting to the Lung Disease Department in order to diagnose a lung tumor (ICD-10 code: D38) in the 12 months prior to the outbreak of the pandemic and 12 months after its start. Our main objective was to collect descriptive data on patients admitted in the two periods (before and during the pandemic) and to assess the combined effect of all components of the pandemic on lung cancer patients.

## 2. Materials and Methods

Our study included patients admitted to the 2nd Department of Lung Diseases, Medical University of Białystok, Poland, for the diagnosis of a lung tumour (ICD-10: D38) 12 months prior to the COVID-19 epidemic and 12 months after the outbreak. Patients admitted from 1 April 2019 to 31 March 2020 were considered the pre-COVID-19 group and patients admitted in the period from 1 April 2020 to 31 March 2021 were considered the COVID-19 group, since the outbreak of the epidemic in Poland began in according time period (the first case of COVID-19 in Poland was on 4 March 2020, and the first case of COVID-19 in Białystok where the study was performed was diagnosed on the 17 March 2020). The Medical University of Bialystok Clinical Hospital, where the study was performed, is the largest and most-advanced healthcare operator in north-eastern Poland, with over 50,000 hospitalizations and 200,000 outpatient appointments annually. It was the largest hospital receiving and curing COVID-19 patients in this part of Poland in the described period. Patients with metastatic lung tumours originating from other organs and benign tumours were not included in the analysis to limit the analysis to confirmed lung cancer cases only. The collected data contained the information presented in Table 1. The data was then categorized for further analysis according to Table 2.

Patient data comes from analyzing electronic medical records stored in the hospital’s electronic documentation system, CliniNET. CliniNET is an electronic data management (EDM) system used to describe the patient’s history and findings in examination and administering treatment. It collects and stores all the results significant for patient care, including laboratory tests, imaging and histopathology.

The type of treatment used was divided into two groups according to the radicalness, assuming as a radical treatment following options: surgical treatment, radical chemoradiotherapy, and radical radiotherapy, whereas the other treatment methods were assumed as non-radical treatment.

Because of the lack of data on the exact date of death, in order to indirectly assess the impact of COVID-19 on the treatment of patients, we used the available data on the patient’s dead/alive status in the eWUŚ (national public electronic system enabling immediate confirmation of the patient’s right to health care services) database as of 25 November 2021, and the date of the last contact with the patient regarding a hospital visit, outpatient clinic visit, or telephone contact. Death-relating dates were included only if the patient’s death occurred in the hospital. Combining the information from these two sources allowed us to make an indirect assessment of survival, which we performed using the Kaplan–Meier analysis.

All the statistics were performed with STATISTICA 13.0 software (StatSoft, Kraków, Poland). Normality was assessed using the Shapiro–Wilk test. Comparison between groups was performed using the Chi-squared or Mann–Whitney U test when applicable. The survival rate was assessed using Kaplan–Meier curves. Significance was assessed with the log-rank test. In all cases, *p* < 0.05 was considered statistically significant.

## 3. Results

We managed to include 320 people who met the inclusion criteria. Clinical characteristics of the patients are presented in Table 3.

Most patients were 65–74 years (*n* = 143), 98 were aged <65 years, and 79 were 75 years or older. The mean age and median for the pre-COVID-19 and COVID-19 groups were 69.59 (Me: 70) and 67.66 (Me: 69) years, respectively. There was no statistically significant difference in median age between groups (pre-COVID-19 vs. COVID-19 *p* = 0.057).

The stage of disease was determined in accordance with the latest 8th edition of the TNM classification. Although the TNM staging system is generally not as important for SCLC as it is for NSCLC, it can be used for both malignancies.

For further analysis, patients were divided into categories regarding clinical stage (CS) and ECOG PS, as shown in Table 4.

Data on the treatment method used were reported (chemotherapy *n* = 173, radical radiotherapy *n* = 7, surgery *n* = 28, symptomatic treatment *n* = 26, chemoradiotherapy *n* = 27, palliative radiotherapy *n* = 4, no treatment due to death *n* = 8, immunotherapy *n* = 28, no treatment required *n* = 19). The type of treatment used is presented in the table below (Table 5).

In our study, we found a statistically significant difference in the performance status according to the WHO/ECOG PS scale in patients admitted to the hospital for the diagnosis of a lung tumour—in the pre-COVID-19 period, a more significant percentage of patients had high-performance status assessed as WHO 0-1 compared to the corresponding period in the COVID-19 period (ECOG PS 0-1 pre-COVID-19 vs. COVID-19: 107 (81.1%) vs. 135 (71.8%); ECOG PS ≥ 2 pre-COVID-19 vs. COVID-19: 24 (18.2%) vs. 53 (28.2%); *p* = 0.04). A similar relationship could be seen after the analysis of the clinical advancement of the disease. CS assessed on admission in the pre-COVID-19 and COVID-19 periods—during the pandemic, the percentage of patients presenting with more advanced cancer was significantly higher (CS I-IIIA pre-COVID-19 vs. COVID-19: 28 (21.2%) vs. 22 (11.7%); CS IIIB-IV pre-COVID-19 vs. COVID-19: 104 (78.8) vs. 165 (87.7). In comparison of the type of treatment used in the pre-COVID-19 and COVID-19 periods, we can observe that in the COVID-19 period, there were more cases of immunotherapy use and more patients who did not start the treatment because of death. We observed no significant difference in the frequency of other treatments used. We also compared the pre-COVID-19 and COVID-19 groups in terms of the percentage of radical treatment used—based on the data obtained, no significant difference in the frequency of radical treatment used in the period before and after the pandemic was found (radical treatment pre-COVID-19 vs. COVID-19: 29 (22%) vs. 32 non-radical treatment pre-COVID-19 vs. COVID-19 *p* = 0.5).

In a further analysis, we also performed a survival analysis between the pre-COVID-19 and COVID-19 groups. In the analysis, we mainly used the date of the last contact due to the inability to obtain information on the death date. There was no statistically significant difference in overall survival between the COVID-19 and pre-COVID-19 groups based (log-rank *p* = 0.63) (Figure 1).

As expected, patients treated radically were characterized by significantly longer survival (log-rank *p* = 0.0001) (Figure 2). However, there was no significant difference in survival between the analogous groups (radically and non-radical treated) in the pre-COVID-19 and COVID-19 periods (*p* = 0.89 and *p* = 0.6, respectively) (Figure 3 and Figure 4).

Interesting results were observed when comparing the survival in individual histological types between the pre-COVID-19 and COVID-19 groups. After dividing into particular types, a significant increase in the survival of patients with adenocarcinoma during the pandemic period was observed compared to the period prior to the pandemic (log-rank *p* = 0.07) (Figure 5).

On the other hand, a significant reduction in the survival time of patients with NOS carcinoma during the pandemic compared to the pre-pandemic was observed (log-rank *p* = 0.04) (Figure 6).

No statistically significant differences in survival were found in the remaining histological types (squamous and small-cell lung cancer) (Figure 7 and Figure 8).

## 4. Discussion

The coronavirus SARS-CoV-2 (COVID-19) outbreak affected several aspects of the management of patients with cancer. Initially, the impact of the pandemic on the oncological population was difficult to estimate, and the factors responsible for the consequences were the impact of the infection itself, as well as the indirect effects resulting from the collapse of the healthcare system and the general economic crisis [18]. As shown in COVID-19 and the cancer Research Network analysis (CCRN), trends suggested a significant decrease in all cancer-related patient encounters caused by the pandemic [19].

In a study by Rodriguez et al., conducted on the US population in the early stages of the pandemic, nearly half of the patients experienced changes in care because of the impact of the pandemic. Notably, the impact of the pandemic on cancer-related concerns, as well as the impact on social determinants of health, was more severe in people with advanced disease [20].

It was observed that patients with simultaneous cancer and COVID-19 had a higher risk of a severe disease course than patients without comorbid cancer [21]. Cancer patients had higher mortality rates from COVID-19 [22,23], especially those with lung cancer [24,25]. 

Treatment disorders and delays in cancer diagnosis may be associated with COVID-19, which may translate into negative clinical consequences in this group of patients. Many cancer patients have experienced changes in their lung cancer treatment plans [26]. COVID-19 reduced the number of lung cancer patients receiving cancer treatment. In the study by Araujo et al. in the Latin American Medical centre, two intervals were compared: pre-COVID-19 (March to May 2019) and COVID-19 pandemic (March to May 2020) periods. A significant decrease in patients undergoing cancer treatment was observed after the COVID-19 pandemic [27].

Reyes et al. [28] found that 38% fewer new lung cancer cases were diagnosed during COVID-19 compared with pre-COVID-19, with more symptomatic and severe NSCLC diagnosed during the pandemic. In the same study, the 30-day mortality rate in patients with newly diagnosed NSCLC significantly increased with the pandemic from 25% before COVID-19 to 49% during COVID-19. With SCLC’s new diagnoses, the 30-day mortality rate jumped from 18% pre-COVID-19 to 32% during COVID-19. The observations are aligned with results obtained in our study. The ongoing study (MA03.08) revealed that lung cancer diagnosis was affected during the COVID-19 pandemic, with fewer cases diagnosed and more symptomatic diseases compared to 2019, which seems to be associated with worse outcomes [26]. 

The mortality rate in our study was not assessed because patient death occurring outside of the hospital is not reported. For our study, we used the date of the last contact with the patient and information about the dead or alive status via the eWUŚ system, but the calculations based on these data are of limited value. They do not accurately reflect the survival time because the loss of contact could have been influenced by either the patient’s death and qualification for palliative treatment or hospitalization and treatment in another hospital.

An interesting fact in our study is the significantly longer survival of patients with adenocarcinoma during a pandemic compared to the period before the pandemic. One of the possible reasons for this may be the increased recruitment of these patients to clinical trials caused by the introduction of NGS as the standard of care in our centre for each patient diagnosed with adenocarcinoma, which may also explain increased immunotherapy use in our department in the COVID-19 era.

## 5. Conclusions

The COVID-19 pandemic has had an early negative effect on lung cancer patients. As shown, during the COVID-19 period, patients reporting lung tumours tended to have worse performance status upon admission. Patients were admitted with the more advanced disease during the pandemic time. More immunotherapy and more deaths before the start of treatment in the COVID-19 period occurred during the COVID-19 period. The general OS was similar between the two periods. However, we observed longer OS in adenocarcinoma patients and shorter OS in NOS cancer patients during COVID-19. Further studies focusing on the analysis of the causes of differences between the groups are needed.

## Figures and Tables

**Figure 1 ijerph-20-01583-f001:**
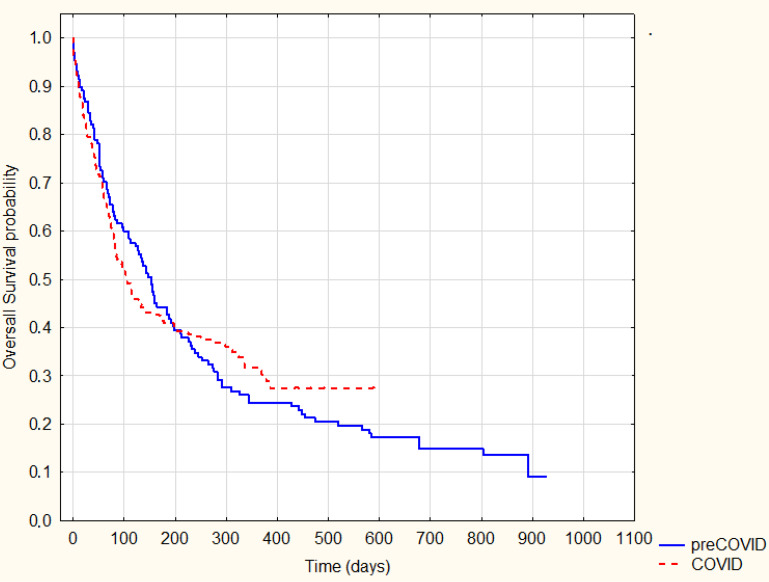
Kaplan–Meier Survival Analysis—pre-COVID-19 vs. COVID-19 overall survival, log-rank. *P* = 0.63.

**Figure 2 ijerph-20-01583-f002:**
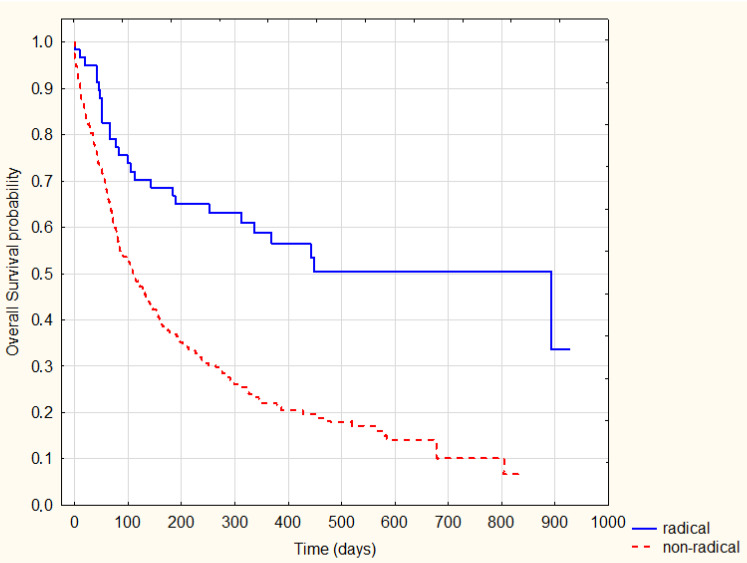
Kaplan–Meier Survival Analysis—radical vs. non-radical treatment, log-rank *p* = 0.0001.

**Figure 3 ijerph-20-01583-f003:**
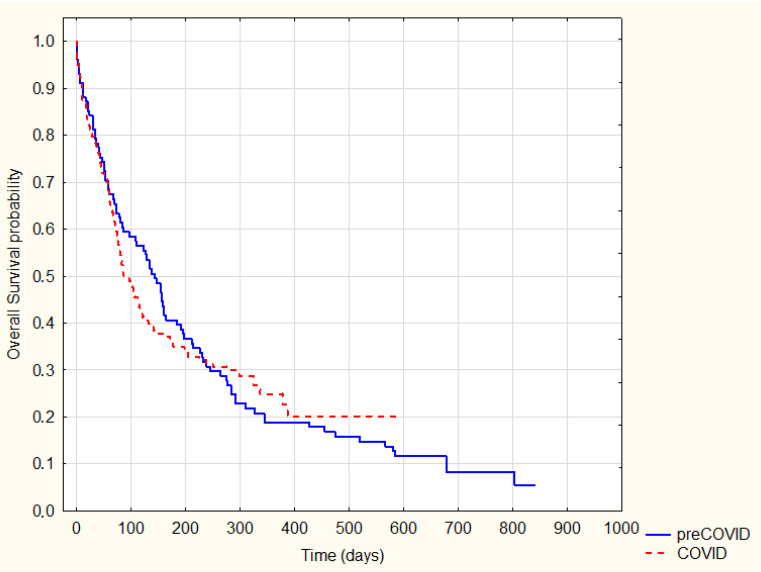
Kaplan–Meier Survival Analysis—non-radical treatment pre-COVID-19 vs. COVID-19, log-rank *p* = 0.89.

**Figure 4 ijerph-20-01583-f004:**
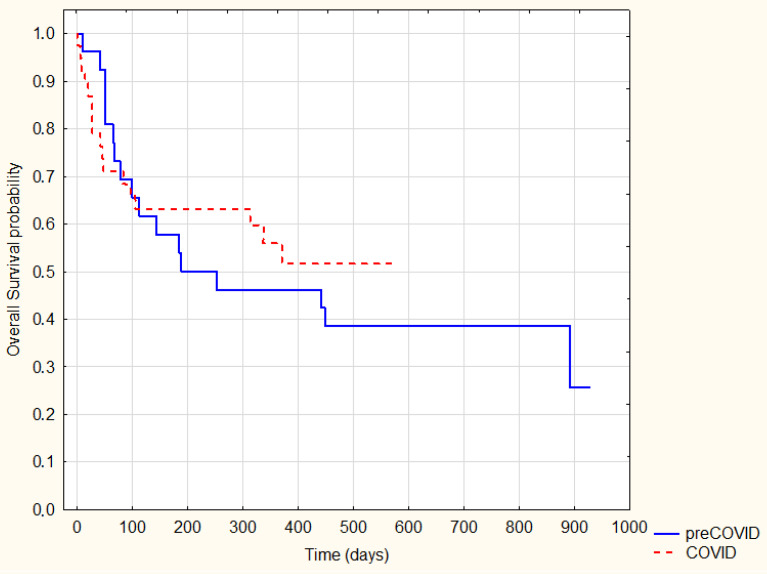
Kaplan–Meier Survival Analysis—radical treatment pre-COVID-19 vs. COVID-19, log-rank *p* = 0.6.

**Figure 5 ijerph-20-01583-f005:**
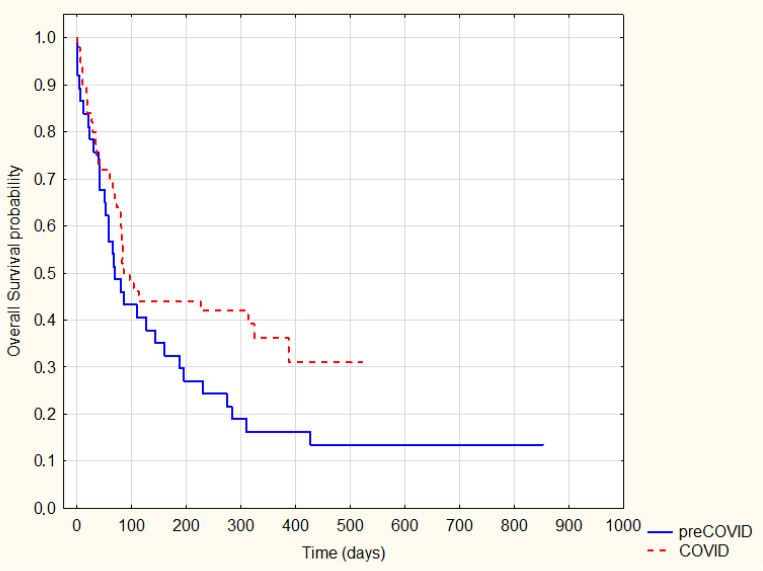
Kaplan–Meier Survival Analysis—adenocarcinoma pre-COVID-19 vs. COVID-19, log-rank *p* = 0.07.

**Figure 6 ijerph-20-01583-f006:**
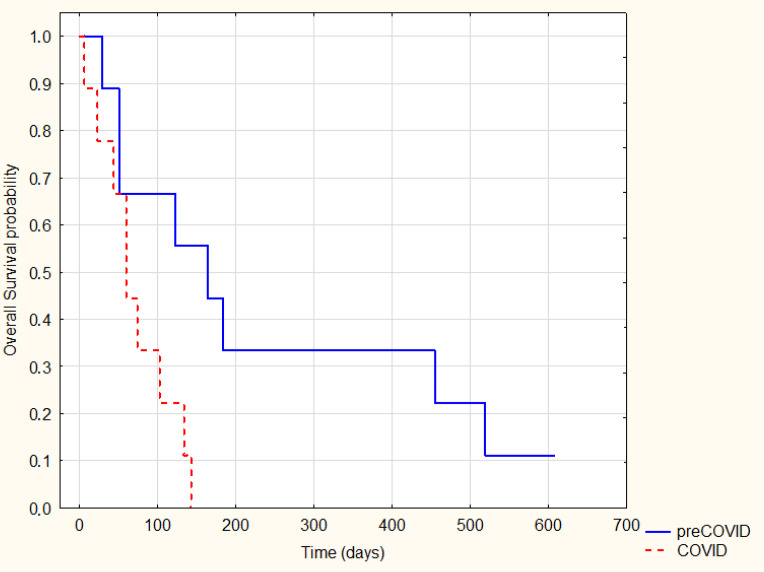
Kaplan–Meier Survival Analysis—NOS cancer pre-COVID-19 vs. COVID-19, log-rank *p* = 0.04.

**Figure 7 ijerph-20-01583-f007:**
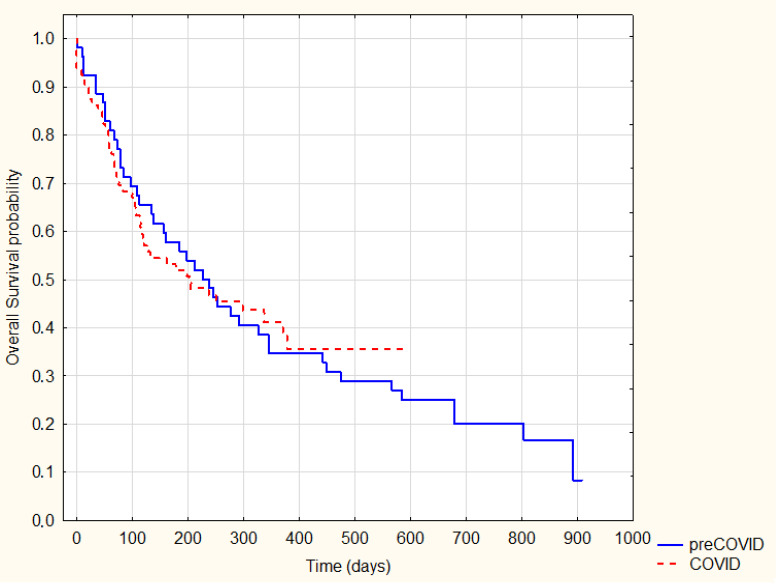
Kaplan–Meier Survival Analysis—squamous cancer pre-COVID-19 vs. COVID-19, log-rank *p* = 0.83.

**Figure 8 ijerph-20-01583-f008:**
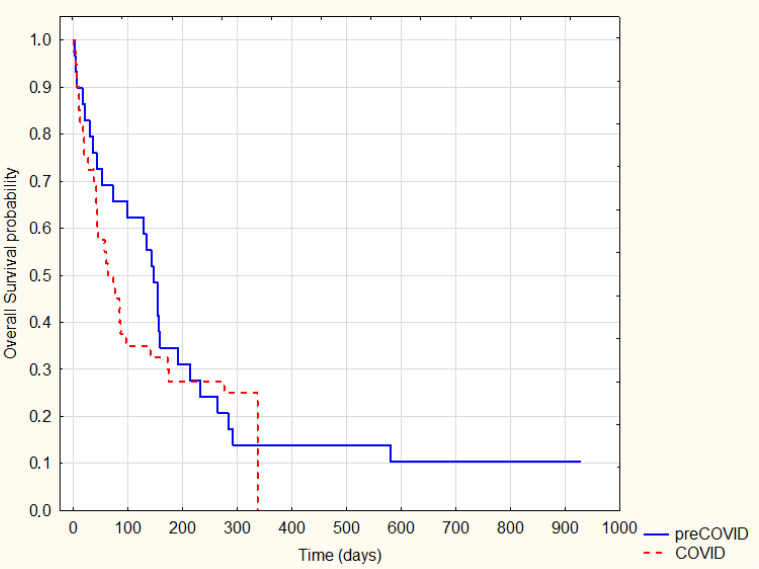
Kaplan–Meier Survival Analysis—small cell lung cancer pre-COVID-19 vs. COVID-19, log-rank *p* = 0.71.

**Table 1 ijerph-20-01583-t001:** Data collected via retrospective analysis.

Collected Data
Age
Sex
Date of admission
Date of diagnosis
Histopathology
Clinical stage (CS)
ECOG PS
Treatment administered Radicalness of administered treatment
Date of death, if available, or date of the last contact with the patient

**Table 2 ijerph-20-01583-t002:** The categories used in the analysis.

Parameter	Categories
Age	<65
	65–74
	≥75
Clinical stage (CS)	Non-advanced lung cancer (I-IIIA)
	Advanced lung cancer (IIIB-IV)
Radicalness of treatment	no radical treatment (palliative chemotherapy, palliative radiotherapy or immunotherapy, best supportive care)
	radical treatment (surgery, radical radiotherapy, radical radiochemotherapy of NSCLC)
	No information available
ECOG Performance Status	Good (ECOG PS 0-1)
	Poor (ECOG PS 2-4)

**Table 3 ijerph-20-01583-t003:** The clinical characteristics of patients included in the analysis.

Clinical Characteristic	Total, *n* = 320 (%)	Pre-COVID-19, *n* = 132	COVID-19, *n* = 188
Sex, *n* (%)			
Male	229 (71.6)	97 (73.5)	134 (71.3)
Female	91 (28.4)	35 (26.5)	54 (28.7)
Age (years), median	68,46 (Me: 69)	69.59 (Me: 70)	67.66 (Me: 69)
Histopathology, *n* (%)			
Squamous cell carcinoma	135 (42.2)	54 (40.9)	81 (43.1)
Adenocarcinoma	87 (27.2)	37 (28.0)	50 (26.6)
Small cell lung cancer	71 (22.2)	30 (22.7)	41 (21.8)
NOS	19 (5.9)	10 (7.6)	9 (4.8)
No diagnosis	8 (2.5)	1 (0.8)	7 (3.7)
CS category, *n* (%)			
IIA	17 (5.3)	14 (10.6)	3 (1.6)
IIB	9 (2.8)	4 (3.0)	5 (2.7)
IIIA	25 (7.8)	10 (7.6)	15 (8.0)
IIIB	33 (10.3)	10 (7.6)	23 (12.2)
IIIC	3 (0.9)	0 (0.0)	3 (1.6)
IV	232 (72.5)	94 (71.2)	138 (73.4)
No data	1 (0.3)	0 (0.0)	1 (0.5)
ECOG Performance Status, *n* (%)			
0	14 (4.4)	10 (7.6)	4 (2.1)
1	227 (70.9)	97 (73.5)	130 (69.1)
2	40 (12.5)	7 (5.3)	33 (17.6)
3	11 (3.4)	6 (4.5)	5 (2.7)
4	26 (8.1)	11 (8.3)	15 (8.0)
No data	2 (0.6)	1 (0.8)	1 (0.5)

**Table 4 ijerph-20-01583-t004:** Patients divided into categories by clinical stage and ECOG performance status.

Category	Total, *n* = 320	Pre-COVID-19, *n* = 132	COVID-19, *n* = 188
Clinical stage (CS), *n* (%)			
Non-advanced lung cancer (I-IIIA)	51 (15.9)	28 (21.2)	22 (11.7)
Advanced lung cancer (IIIB-IV)	268 (83.8)	104 (78.8)	165 (87.7)
No information	1 (0.3)	0 (0.0)	1 (0.5)
ECOG PS, *n* (%)			
Good (ECOG PS 0-1)	242 (75.6)	107 (81.1)	135 (71.8)
Poor (ECOG PS 2-4)	77 (24.1)	24 (18.2)	53 (28.2)
No information	1 (0.3)	1 (0.8)	0 (0.0)

**Table 5 ijerph-20-01583-t005:** Comparison of types of treatment used in pre-COVID-19 and COVID-19 periods.

Type of Treatment	Total	Pre-COVID-19	COVID-19	*p*
CHTX	173	79	94	0.18
RTX	7	4	3	0.7
Surgery	28	17	11	0.25
Palliative	26	14	12	0.69
RTX and CHTX	27	9	18	0.08
Non-radical RTX	4	1	3	0.32
Death before treatment administered	8	1	7	0.03 *
Immunotherapy	28	4	24	0.001 *

* *p* <0.05 CHTX—chemotherapy, RTX—radiotherapy.

## Data Availability

Data available on request from the authors.

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
