# Peer review of "Impact of COVID-19 in Patients with Lung Cancer: A Descriptive Analysis"

_ijerph, 2023, doi:10.3390/ijerph20021583_

Round 1
Reviewer 1 Report
This study is a comparison of lung cancer related outcomes before and during the Covid pandemic. The authors conclude a few outcomes differed between the periods and most did not.
Sections of this paper are not well written making it very difficult to understand. The methods section, in particular, is very brief and lacking important details which would help readers understand what was done and determine the validity of the findings and conclusions.
I found it difficult to discern when the authors were considering covid related problems in treatment, access to care, or individual-level infection. For example, line 159, “Treatment disorders and delays in cancer diagnosis may be associated with COVID19, which may translate into negative clinical consequences in this group of patients.” They simply compared the two periods, before and during the pandemic, but they could have improved this paper if they better explained the mechanisms of concern with various outcomes investigated.
The Methods section is so brief it makes it difficult or impossible to be sure the Results are valid. In the Methods section they merely describe the data source, the CliniNET electronic records system. This source needs to be better described. They mention the location in the Abstract yet fail to describe even this in the methods section.
There is no mention of statistical analysis in the Methods whatsoever. In the Results section, the authors note “In further analysis, we also performed a survival analysis between the preCOVID and COVID groups.” They show Kaplan-Meier graphs and report p-values. But, again, there’s no mention of the application of this technique in the Methods section. I see no indication of how long a window was used to examine deaths among their subjects. They do not indicate the unit of time on their K-M graphs. The time durations vary from 1100 to 700 along the x-axes. This confusion is exacerbated as they report looking at 3-month pre-Covid and Covid period windows in the Abstract but 12-month window in the Introduction. I cannot tell what the statistical analysis entailed.
In the Results they list several descriptive findings which belong in a table. This would shorten the Results.
The authors report “In our study, we found a statistically significant difference in…” but there’s no mention of what technique they used to test for differences.
Line 82. The authors report what should be found in the Methods in their Results section: “The type of treatment used was divided into two groups according to the radicalness, assuming as a radical treatment following options: surgical treatment, radical chemoradiotherapy, and radical radiotherapy, while the other treatment methods were assumed as non-radical treatment (non-radical treatment n = 259, radical treatment n = 61).” This sort of information belongs in the Methods section.
The limitations they report regarding the censoring of patients in their Kaplan-Meier results (which appear to be the main findings) may be so strong as to make conclusions unsound. This difficulty might be seen as not such a problem if the methods were presented more clearly. It is hard to interpret findings without more detail.
Author Response
Dear Sir or Madam,
Thank you for your time and insightful comments on the paper. We appreciate the time and effort you have dedicated to provide your valuable feedback.We have been able to incorporate changes to reflect all of the suggestions provided. The language and grammar were overall checked and improved. We have highlighted in red the changes within the manuscript.
Here is a point-by-point response to comments and concerns:
- Comment 1: The methods section, in particular, is very brief and lacking important details which would help readers understand what was done and determine the validity of the findings and conclusions.
- The methods section was restructured and supplemented with missing information regarding the collection of data and statistics
- The methods section was restructured and supplemented with missing information regarding the collection of data and statistics
- Comment 2: I found it difficult to discern when the authors were considering covid related problems in treatment, access to care, or individual-level infection. For example, line 159, “Treatment disorders and delays in cancer diagnosis may be associated with COVID19, which may translate into negative clinical consequences in this group of patients.” They simply compared the two periods, before and during the pandemic, but they could have improved this paper if they better explained the mechanisms of concern with various outcomes investigated.
- Indeed, preliminary conclusions indicate the need for further research focused on the analysis of the causes of differences between the groups – a statement regarding this point was added to the „Conclusions” section. In our study, however, we have not collected data that would allow us to analyze the individual factors affecting patients with lung cancer during COVID-19. Our main objective was to collect descriptive data on patients admitted in the two periods before and during the pandemic and to assess the combined effect of all components of the pandemic expressed as an indirect assessment of survival.
- Indeed, preliminary conclusions indicate the need for further research focused on the analysis of the causes of differences between the groups – a statement regarding this point was added to the „Conclusions” section. In our study, however, we have not collected data that would allow us to analyze the individual factors affecting patients with lung cancer during COVID-19. Our main objective was to collect descriptive data on patients admitted in the two periods before and during the pandemic and to assess the combined effect of all components of the pandemic expressed as an indirect assessment of survival.
- Comment 3: The Methods section is so brief it makes it difficult or impossible to be sure the Results are valid.
- The methods section was restructured and supplemented with missing information regarding the collection of data and statistics
- The methods section was restructured and supplemented with missing information regarding the collection of data and statistics
- Comment 4: In the Methods section they merely describe the data source, the CliniNET electronic records system. This source needs to be better described.
- The information regarding the data source CliniNET was added to the methods section
- The information regarding the data source CliniNET was added to the methods section
- Comment 5: They mention the location in the Abstract yet fail to describe even this in the methods section.
- The information regarding the location and hospital was added to the methods section.
- The information regarding the location and hospital was added to the methods section.
- Comment 6: There is no mention of statistical analysis in the Methods whatsoever
- The statistical analysis description was added to the „Methods” section
- The statistical analysis description was added to the „Methods” section
- Comment 7: In the Results section, the authors note “In further analysis, we also performed a survival analysis between the preCOVID and COVID groups.” They show Kaplan-Meier graphs and report p-values. But, again, there’s no mention of the application of this technique in the Methods section
- The statistical analysis description was added to the „Methods” section
- The statistical analysis description was added to the „Methods” section
- Comment 8: Results, Kaplan Meier analysis: I see no indication of how long a window was used to examine deaths among their subjects.
- Dead/alive status for all of the patients (preCOVID and COVID) was assessed on 25.11.2021, information was added to the „Methods” section
- Dead/alive status for all of the patients (preCOVID and COVID) was assessed on 25.11.2021, information was added to the „Methods” section
- Comment 9: They do not indicate the unit of time on their K-M graphs.
- The unit of time was added to all K-M graphs
- The unit of time was added to all K-M graphs
- Comment 10: The time durations vary from 1100 to 700 along the x-axes.
- The time duration scale along the x-axes was unified between all the Kaplan-Meier curves beside the NOS OS curve (survival was shorter here and therefore scale 700 days was left due to clarity and esthetic reasons
- The time duration scale along the x-axes was unified between all the Kaplan-Meier curves beside the NOS OS curve (survival was shorter here and therefore scale 700 days was left due to clarity and esthetic reasons
- Comment 11: This confusion is exacerbated as they report looking at 3-month pre-Covid and Covid period windows in the Abstract but 12-month window in the Introduction. I cannot tell what the statistical analysis entailed.
- The error in the abstract was fixed – pre-COVID and COVID period windows should be both 12 months
- The error in the abstract was fixed – pre-COVID and COVID period windows should be both 12 months
- Comment 12: In the Results they list several descriptive findings which belong in a table. This would shorten the Results.
- Descriptive findings were converted into a table and presented as such in the „Results” section
- Descriptive findings were converted into a table and presented as such in the „Results” section
- Comment 13: The authors report “In our study, we found a statistically significant difference in…” but there’s no mention of what technique they used to test for differences.
- Information regarding techniques used to test for differences between the groups was added in the „Methods” section in the statistical analysis description
- Information regarding techniques used to test for differences between the groups was added in the „Methods” section in the statistical analysis description
- Comment 14: Line 82. The authors report what should be found in the Methods in their Results section: “The type of treatment used was divided into two groups according to the radicalness, assuming as a radical treatment following options: surgical treatment, radical chemoradiotherapy, and radical radiotherapy, while the other treatment methods were assumed as non-radical treatment (non-radical treatment n = 259, radical treatment n = 61).” This sort of information belongs in the Methods section.
- The „Type of treatment section” has been moved from the Results section to the Methods section
- The „Type of treatment section” has been moved from the Results section to the Methods section
- Comment 15: The limitations they report regarding the censoring of patients in their Kaplan-Meier results (which appear to be the main findings) may be so strong as to make conclusions unsound. This difficulty might be seen as not such a problem if the methods were presented more clearly. It is hard to interpret findings without more detail.
- The methods section was restructured and supplemented with missing information regarding the collection of data and statistics
Reviewer 2 Report
I do not understand why it is put as ICD-10 code D38, when, according to ICD-10, Malignant neoplasm of bronchus and lung has code C34 (https://icd.who.int/browse10/2010/en#/C34). According to ICD_10, code D indicates Benign neoplasms.
It is not clear from the Introduction in which country the analysis was carried out and this information is not deductible in the Materials and Methods section either.
Results: A statistical test of the differences between the two populations' average ages should be added.
The part about staging should be put in the methods, not in the results.
In the discussion, it is reported, "The mortality rate in our study was not assessed because patient death occurring outside of the hospital is not reported". It should therefore be explained better how the survival analysis was done.
Author Response
Dear Sir or Madam,
Thank you for your time and insightful comments on the paper. We appreciate the time and effort you have dedicated to providing your valuable feedback. We have been able to incorporate changes to reflect all of the suggestions provided. The language and grammar were overall checked and improved. We have highlighted in red the changes within the manuscript.
Here is a point-by-point response to comments and concerns:
- Comment 1: I do not understand why it is put as ICD-10 code D38, when, according to ICD-10, Malignant neoplasm of bronchus and lung has code C34 (https://icd.who.int/browse10/2010/en#/C34). According to ICD_10, code D indicates Benign neoplasms.
- The ICD D38 code is used at the beginning of diagnostics in patients with lung tumor with no histopathological confirmation; the C34 ICD-10 code is used after obtaining histopathological diagnosis which may take various amount of time and hence the use of D38 code in our paper to track the time since the very beginning of diagnostics.
- Comment 2: It is not clear from the Introduction in which country the analysis was carried out and this information is not deductible in the Materials and Methods section either.
- The information regarding the location and hospital was added to the methods section.
- The information regarding the location and hospital was added to the methods section.
- Comment 3: Results: A statistical test of the differences between the two populations' average ages should be added.
- There was no statistically significant difference in median age between groups, preCOVID vs COVID p=0,057; information added to the manuscript in according section in „Results”
- There was no statistically significant difference in median age between groups, preCOVID vs COVID p=0,057; information added to the manuscript in according section in „Results”
- Comment 4: The part about staging should be put in the methods, not in the results.
- The „Type of treatment section” has been moved from the Results section to the Methods section
- The „Type of treatment section” has been moved from the Results section to the Methods section
- Comment 5: In the discussion, it is reported, "The mortality rate in our study was not assessed because patient death occurring outside of the hospital is not reported". It should therefore be explained better how the survival analysis was done.
- The methods section was restructured and supplemented with missing information regarding the collection of data and statistics
Round 2
Reviewer 2 Report
The work is overall improved, the introduction should be expanded as it does not provide sufficient background and does not include all relevant references.
Author Response
Dear Sir or Madam,
Once again thank you for your time and insightful comments on the paper. We have been able to incorporate changes to reflect all of the suggestions provided. As previously, we have highlighted in red the changes within the manuscript
Below we attach the response to the comment:
- Comment 1: The work is overall improved, the introduction should be expanded as it does not provide sufficient background and does not include all relevant references.
- The introduction section has been expanded and supplemented with corresponding references relevant to the subject.